# Assessment of the diagnostic performance of the SD Bioline Malaria antigen test for the diagnosis of malaria in the Tombel health district, Southwest region of Cameroon

Abigail L. Ngalame[1‡], Djakissam Watching[2‡*], Odette Z. Kibu[1‡], Elisabeth M. Zeukóo[3‡], Dickson S. Nsagha[1‡*]

1 Department of Public Health and Hygiene, Faculty of Health Sciences, University of Buea, Buea, Cameroon, 2 Department of Medical Laboratory Sciences, Faculty of Health Sciences, University of Buea, Buea, Cameroon, 3 Department of Biomedical Sciences, Faculty of Health Sciences, University of Buea, Buea, Cameroon

‡ ALN, DW and DSN are contributed equally to this work. OZK and EMZ are also contributed equally to this work.
* nsaghads@hotmail.com (DSN); djakissamwatching@yahoo.fr (WD)

## Abstract

Malaria rapid diagnostic tests (mRDTs) are commonly used for the diagnosis of malaria in resource-limited settings. However, the sensitivity of RDTs may vary depending on the brand. The aim of this study was to assess the diagnostic performance the SD Bioline Malaria *Plasmodium falciparum* antigen rapid diagnostic test (Ag *P.f* RDT) for the diagnosis of malaria in the Tombel Health District (THD). This was a cross-sectional community-based study targeting persons living within the THD from 30th April to 20th June 2023. A consecutive sampling technique was used to collect blood samples among 250 symptomatic and asymptomatic individuals and tested using the SD Bioline Malaria Ag *P.f* RDT and microscopy. Data was analysed using SPSS version 25.0. The sensitivity, specificity, positive predictive value (PPV) and negative predictive value (NPV) were calculated to assess the diagnostic accuracy of the RDT. Microscopy identified 133 (53.2%) symptomatic and asymptomatic participants with malaria and the mRDT identified 60 (24%) cases with 52 (20.8%) low parasites density and 8 (3.2%) moderate parasites density. The sensitivity of Malaria Ag *P.f* RDT was 45.0%, the test correctly identified 60 (45%) of true positive *P. falciparium* cases. The specificity of 100% showed that the test correctly identified all negative cases. PPV was 100%, there is a very high likelihood that all the positive cases are truly infected. The lower NPV of 61.5%, there is still a chance the negative cases might be infected. The Area Under the Curver (AUC) = 0.78 with moderate diagnostic suggesting that there is 22% chance that the test can produce a false positive result. The overall accuracy of mRDT in the THA is moderate. This level of accuracy may be acceptable in some contexts, but it is not ideal for a screening test, particularly for a disease like malaria that can have serious health consequences if left untreated.

**Data availability statement:** All relevant data are within the paper and its Supporting Information files.

**Funding:** The author(s) received no specific funding for this work.

**Competing interests:** The authors have declared that no competing interests exist.

## Introduction

Malaria continues to be a serious public health concern. The World Health Organization (WHO) reported 405,000 malaria deaths and 228 million cases in 2018 in all endemic areas [1]. Furthermore, 247 million cases of malaria were reported in 2021 and 245 million cases in 2020, according to the World Malaria Report [2]. In April 2024, the new World Malaria report has mentioned that, there were 608 000 malaria deaths in 2022 in globally. African continent was home to 94% of malaria cases (233 million) and 95% (580 000) of malaria deaths and Over 90% of all malaria deaths occur in sub-Saharan Africa [3]. Malaria accounts for 22% of annual mortality in Cameroon and is the leading cause of hospitalization (46%) and consultation (26%). Over 95% of cases are caused by the primary pathogen, *Plasmodium falciparum* [4]. The risk of malaria transmission remains high throughout the whole country of Cameroon, making it a severe public health concern.

The most potent antimalarial drug is artemisinin-based combination therapy (ACTs), WHO advises treating a patient only if a test for malaria is positive, since there won't be a viable substitute for artemisinin derivatives for a number of years. This is necessary to maintain the effectiveness of ACTs. Effective malaria management requires a timely and precise diagnosis. In Cameroon, clinical assessment, microscopy, and mRDTs are the main methods used to diagnose malaria [5]. Microscopy has significant limitations that prevent its use in resource-constrained nations like Cameroon, where the prevalence of malaria is highest. These include a turnaround time, an unstable power supply, and the requirement for technical competence. Consequently, mRDT was created to increase the objectivity and sensitivity of diagnosing malaria without entirely depending on microscopy [6]. Rapid diagnostic test performance, however, can be impacted by a number of factors, such as the product's format and type, storage conditions, short shelf life due to temperature instability, particularly during transportation from the manufacturing or supply location to the final place of use, suboptimal sensitivity at low parasite densities, and the inability to identify the species of the parasite or quantify its density [7,8,9,10]. Inaccurate diagnostic testing can result in incorrect therapy, raising the possibility of morbidity and death [11]. Consequently, a cross-sectional community-based investigation is required to evaluate the diagnostic performance of these mRDT for the diagnosis of malaria in rural populations. This research will yield important data regarding the precision of the RDT utilized in the THA as well as its potential benefits for enhancing malaria diagnosis and treatment in rural settings [12]. Therefore, this study was designed to assess the diagnostic performance of the SD BIOLINE malaria Antigen (Ag) *Plasmodium falciparum* (*P.f*) test for the diagnosis of malaria in the Tombel Health Area, Southwest region of Cameroon.

## Materials and methods

### Study design and period

This was community based cross-sectional study carried out in the Tombel Health Area in the South West Region of Cameroon.

The protocol was developed, data collected, analysed and interpreted within a period of 6 months (February to July 2023). Data were collected and tested simultaneously all over the investigation period. The administration of questionnaires and collection of samples started on the 30th of April 2023 and ended on the 20th of June 2023.

### Study area and setting

Tombel Health District (THD) in South West Region of Cameroon. is situated in a rural area, with a population of approximately 65000 people with approximately 35,690 males and 28,587

females with about 24 901 children aged 0-16 years. The health area is characterized by dense forests, hilly terrains, and a warm and humid climate.

Despite the presence of health facilities, access to healthcare remains a challenge in THD due to the rugged terrain and poor road network. Additionally, the health facilities face challenges such as inadequate staffing, limited medical supplies, and a lack of equipment and infrastructure.

### Study population

All persons living in the Tombel Health Area during recruitment of participants.

### Inclusion criteria

Persons living in the THD who had not been on antimalarial drugs within the last two weeks at the time of recruitment were included in this study.

### Exclusion criteria

All Persons on antimalarial or medications like doxycycline that affect malaria parasitaemia were excluded in this study.

### Sample size determination

We used the sample size calculation proportions of sensitivity and specificity method of Burderer and collaborators 1996 [13], cited in Educational Notes in Research Methodology and Medical Statistics by Fahim et al., 2019 [14],

$$n = n1 + n2v$$

$$n1 = (TP + FN) / P$$

$$n2 = (TN + FP) / (1 - P)$$

And

$$TP + FN = Z^2 (Sensitivity(1 - Sensitivity)) / W^2$$

$$TN + FP = Z^2 (Specificity(1 - Specificity)) / W^2.$$

Where, $n_1$ and $n_2$ = required number of participants for sensitivity and specificity with estimated sensitivity and specificity of 95% and 99% respectively [15].

TP = True Positive (Positive for malaria with microscopy)
TN = True Negative (Negative for malaria with microscopy)
FN = False Negative (Positive for malaria with microscopy but negative with the mRDT)
FP = False Positive (Negative for malaria with microscopy but positive with the mRDT)
P = Prevalence of malaria from previous study (34.7%) [16].
Z = Standard normal variation at a confidence level of 95% (standard value of 1.96)
W = 0.05 (5% maximum acceptable width of the 95% confidence interval)
Substituting in the equation above,

$$TP + FN = [1.96]^2 (0.95(1 - 0.95)) / [0.05]^2$$

$$TP + FN = 72.99$$

$$TN + FP = [(1.96)]^2 (0.99(1 - 0.99))/[(0.05)]^2$$

$$TN + FP = 15.21$$

$$n1 = 72.99 / 0.347$$

$$n1 = 210.3$$

$$n2 = 15.21 / (1 - 0.347)$$

$$n2 = 23.3$$

$n_1$ required for sensitivity = 211 participants $n_2$ required for specificity = 24 participants (at least n = 235 participants).

Taking into account a 10% non-response rate, our minimum sample size will be 258.5 ≈ 260 participants.

## Sampling techniques

A consecutive sampling technique was used, based on the quarters in the THD. Nine (9) quarters were randomly selected out of the 11 identified quarters. The probability proportionate to size was used to get the number of participants needed from each quarter in the THD as shown in Table 1. Within each quarter, participants were selected consecutively until the expected number of participants allocated for each quarter was attained.

## Study procedure

**Ethical consideration.** Ethical Clearance was obtained from the Institutional Review Board (FHS-IRB reference number: 2023/1992-02/UB/SG/IRB/FHS) of the University of Buea. Administrative authorizations were obtained from the Department of Public Health, Faculty of Health Sciences, South West Regional Delegation of Public Health (SW-RDPH) (reference number: R11/MINISANTE/SWR/RDPH/PS/544/576) and Tombel District Hospital before data collection. Informed consent was obtained from the participants. Participant's information was obtained and kept confidentially by not taking any of their personal information such as names; rather codes were assigned to each participant after data collection and each participant was given equal treatment.

**Table 1. Probability proportionate to size sampling of participants in various quarters.**

| Residence | No (%) | Proportionate sample size |
|---|---|---|
| CDC **Road** | 360 (4.0) | 13 |
| **Ebubu** | 288 (3.2) | 10 |
| **Elom** | 180 (2.0) | 6 |
| **KUPE** | 756 (8.4) | 22 |
| **Peng** | 252 (2.8) | 8 |
| **Mile19** | 540 (6.0) | 16 |
| **Mile20** | 360 (4.0) | 11 |
| **NGAB** | 396 (4.4) | 12 |
| **WARDS** | 5868 (65.2) | 170 |
| **TOTAL** | **900** | **260** |

### Pre analytical phase

**Recruitment of Participants.** Participants were informed about the research and its objectives. Those who accepted to participate in the research, signed the consent forms and answered the questionnaires.

**Administration of questionnaire and Data Capture.** A structured questionnaire was administered to each of the participant after they signed the consent forms for the major participants and assent forms by the parent or guardians for minor participants. Socio-demographic data such as age, gender, occupation, marital status was obtained.

**Data collection.** The data collection was done by using well-structured closed ended questionnaires. Confidentiality of participants was guaranteed assured to the participants and questionnaires were administered in English. These questionnaires were used to get information on the socio-demographic characteristics of the participants as well as the clinical diagnosis. Blood samples were collected by pricking the finger of each participant and using the sample to conduct a mRDT and a thin and thick film for microscopy.

### Laboratory investigations (S6: Malaria laboratory protocol)

**Thick film.** The slides were labelled with the participant's unique identification number. A drop of 6 μL of blood was placed at the centre the slide. The applicator stick was used to spread the drop in a circular motion in spirals (1 cm diameter) starting from the middle of the drop to form a thick and unformed layer. The slides were laid on a smooth surface and allowed to air dry thoroughly at room temperature.

**Thin film.** Approximately 3 μL of blood was placed on a clean labelled slide near its frosted end. Another slide was placed at a 45-degree angle up to the drop, allowing the drop to spread along the contact line of the two slides and a thin layer of even spread was made. The thin smears were allowed to dry and later fixed by immersing them into absolute methanol for a minute.

**Malaria thin and thick film microscopy.** The thick and thin blood films were stained using 10% Giemsa for 15 minutes then rinsed and allowed to air dry. Slides were examined under the microscope using the oil immersion objective lens. A slide was considered positive when at least one trophozoite of the *P. falciparum* parasite was identified. Parasitaemia was determined by counting the number of parasites per 200 white blood cells and assuming that each subject had 8000 white blood cells/μL of blood [17, 18]. The thin films were used to identify the *P. falciparum* present. Samples were placed in a safe box and transported to the health facility for analysis. The thick and thin film were read by a trained microscopist employed at the Tombel District Hospital.

The morphology of the parasite such as the shape, size and staining characteristics was used to distinguish different species of malaria parasite. The number of malaria parasites and specie seen in a microscopic field was reported.

Calculation of parasites densities and the interpretation of results were done using WHO-recommended malaria microscopy standard operating procedure (MM-SOP-09) [19] as: Parasite density= (number of asexual parasite counted)/ (WBC counted) X 8000/μL blood. Parasite densities were estimated according to parasite density levels reported by Kosack et al [20] as low density = 10 to 90 parasites/μl; moderate density = 100 to 1,000 parasites/μl, and high density = > 1,000 parasites/μl.

### Procedure for RDT Test

Approximately 5μL of blood were used to perform the test using the malaria Ag *P.f* RDT kit, following the manufacturer's instructions. The mRDT cassette was label with participant

identification and 5ul of blood was applied to the sample pad on the test strip. Immediately after the blood clotted, the applicator was firmly applied in the circle and two drops of buffer were applied vertically above the circle and left for 10 minutes to migrate after which the results were read. The results were read following the manufacturer's instructions as: if a single pink color appears on the control line "C", the result was considered as negative. If a pink colored band appears on the "C" control line and a distinct pink colored band also appears on Pf line, the result was considered as positive.

The RDT results were obtained in the field by the principal investigator, a certified and experienced laboratory scientist.

## Data management and analysis

The prevalence of malaria was calculated by dividing the number of positive samples by the total sample size for mRDT and microscopy:

$$Malaria\ Prevalence = (number\ of\ asexual\ positive\ samples)/(Total\ number\ of\ participants) \times 100$$

Descriptive statistics was used to summarize the demographic and clinical characteristics of the participants.

The sensitivity, specificity, positive predictive value (PPV) and negative predictive value (NPV) of the SD Bioline malaria antigen RDT was calculated using standard formulas and obtained along with 95% CI (confidence interval).

The diagnostic accuracy of the RDT was assessed using a standard formula and confirmed by the receiver operating characteristic (ROC) curve. The area under the ROC curve (AUC) was analysed to determine the overall accuracy of the mRDT (S1 Data: Diagnostic test calculation). For the AUC, values range from 0.5 to 1.0 where 0.5 was considered to be indicative of chance performance and 1.0 was considered perfect performance (higher values indicate better performance). Logistic regression analysis was performed to determine any demographic or clinical factors that may affect the results of the mRDT. Chi-square test was used to determine association between parasite density and mRDT positivity. The degree of agreement between the mRDT and microscopy was scaled following the Cohen's Kappa analysis; 0.0-0.2 was considered poor, 0.2-0.4 was considered fair, 0.4-0.6 was considered average, 0.6-0.8 was considered good and 0.8-1.0 was considered excellent.

$$Sensitivity = (True\ positive)/(True\ Positive + False\ Negative) \times 100$$

$$Specificity = (False\ positive)/(False\ negative + True\ positive) \times 100$$

$$Positive\ Predictive\ Value\ (PPV) = (True\ positive)/(True\ positive\ False\ Negative) \times 100$$

$$Negative\ Predictive\ Value\ (NPV) = (False\ Positive)/(False\ Negative + True\ Positive) \times 100$$

$$Accuracy = (True\ Positive + False\ Negative)/(True\ Positive + False\ Positive + False\ Negative + True\ Negative) \times 100$$

Were confirmed by MedCalc version 23.0.6 [21] (S4 Data Diagnostic test computed).

Questionnaires from participants were properly checked for completion and stored in a safe cupboard at the end of the day after data was collected. Data was keyed in by the principal investigator and this was done immediately after collection same day eliminating room

for missing data. Data obtained from questionnaires and laboratory tests were entered into Microsoft Excel (S2 Data collection). A copy of the excel file generated at the end of data collection was exported to SPSS version 25.0 for cleaning and analysis (S3 Data analysis). The Chi square test and Pearson correlation were used to determine the association between Malaria parasites density and mRDT positivity. A P-value < 0.05 was considered as statistically significant (S5 Data: SPSS computed). The questionnaires were kept in a locked cupboard if needed for reference.

## Results

### Sociodemographic characteristic of study participants

A total of 250 participants participated in the study and were screened for malaria parasites (Table 2). All of the participants were recruited from the THD. Their age ranged from 2-70 years, with 108(43.2%) of the participants being within the age group 31-45 years with a mean age (±SD) of 31.1 (±21.5) years and median age of 27 years. Among the study participants, 160 (64%) were females and 90 (36%) were males. Also, about 97 (38.8%) of the participants were reported to be Married/Cohabitating and 141 (56.4%) were seen single while 12 of them

**Table 2. Sociodemographic characteristic of study participants.**

| Variables | Category | No (%) |
|---|---|---|
| **Sex** | Male | 90(36.0) |
| | Female | 160 (64.0) |
| | **Total** | **250 (100)** |
| **Age group (Years)** | < 5 | 26 (10.4) |
| | 5-15 | 13 (5.2) |
| | 16-30 | 38 (15.2) |
| | 31-45 | 108 (43.2) |
| | 46-60 | 34 (13.6) |
| | 60+ | 31 (12.4) |
| | **Total** | **250 (100)** |
| **Occupation** | Business | 57 (22.8) |
| | Farmer | 48 (19.2) |
| | Health personnel | 4 (1.6) |
| | House wife | 15 (6.0) |
| | None | 52 (20.8) |
| | Student | 69 (27.6) |
| | Teacher | 5 (2.0) |
| | **Total** | **250 (100)** |
| **Marital Status** | Single | 141 (56.4) |
| | Married | 97 (38.8) |
| | Widowed | 12 (4.8) |
| | **Total** | **250 (100)** |
| **Highest Level of Education** | Primary | 105 (42.0) |
| | Secondary | 72 (28.8) |
| | High school | 31 (12.4) |
| | Tertiary | 14 (5.6) |
| | None | 28 (11.2) |
| | **Total** | **250 (100)** |

were widowed (4.8%). The majority of the participants were students 69(27.6%) followed by business people 57 (22.8%) and the least of them were health personnel 4 (1.6%). Primary level of education was the most frequent level 105 (42.0%) attained by the participants followed by secondary level 72 (28.8%) and least was the tertiary level 14 (5.6%). Twenty-eight (11.2%) of the participants had no formal education. Christians dominated the study participants 221 (84.0%) and Muslims were 29 (11.6) of the total study population. Table 2 shows a summary of the socio-demographic characteristics.

### Prevalence of malaria in the Tombel health district

Sixty participants (24% [95% CI:19 - 29]) out of 250 were found positive for the malaria RDT (Fig 1) while 133 (53.2% [95% CI:47% - 59]) were positive with microscopy (Fig 2). There is a statistically significant difference between the prevalence of malaria using RDT and its prevalence using microscopy with a P-value <0.001 (Table 3).

### Validity of SD BIOLINE to diagnose *P. falciparum* and concurrence with Microscopy

A total of 250 samples were tested both for Microscopy and mRDT. One hundred and thirty-three (133) samples were found positive with microscopy, seventy-three samples among

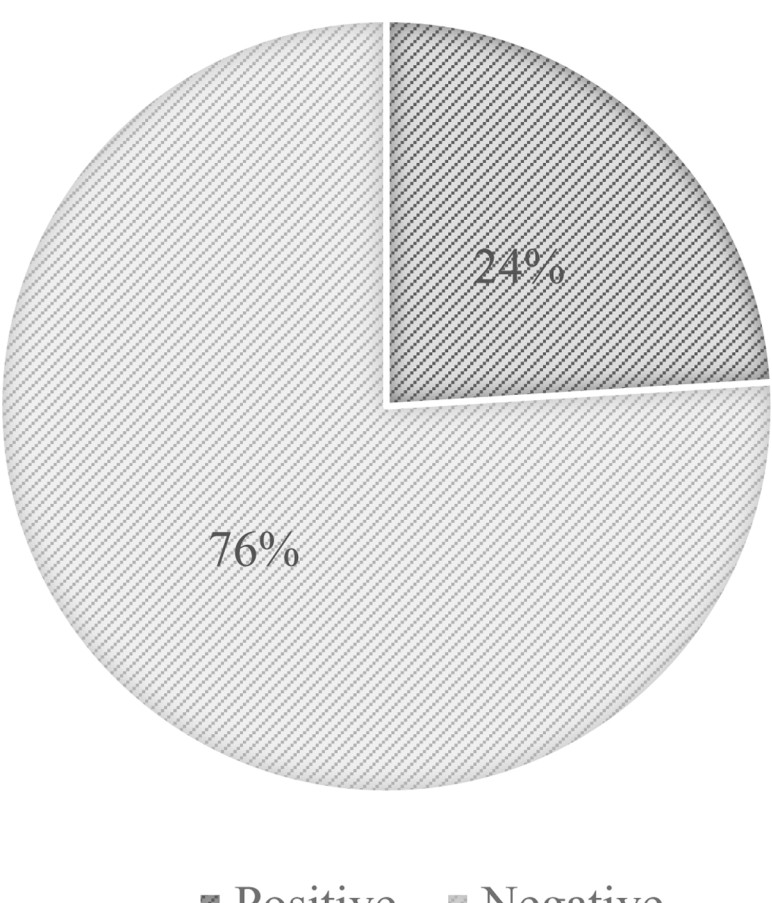

**Fig 1. Prevalence of malaria with mRDT.**

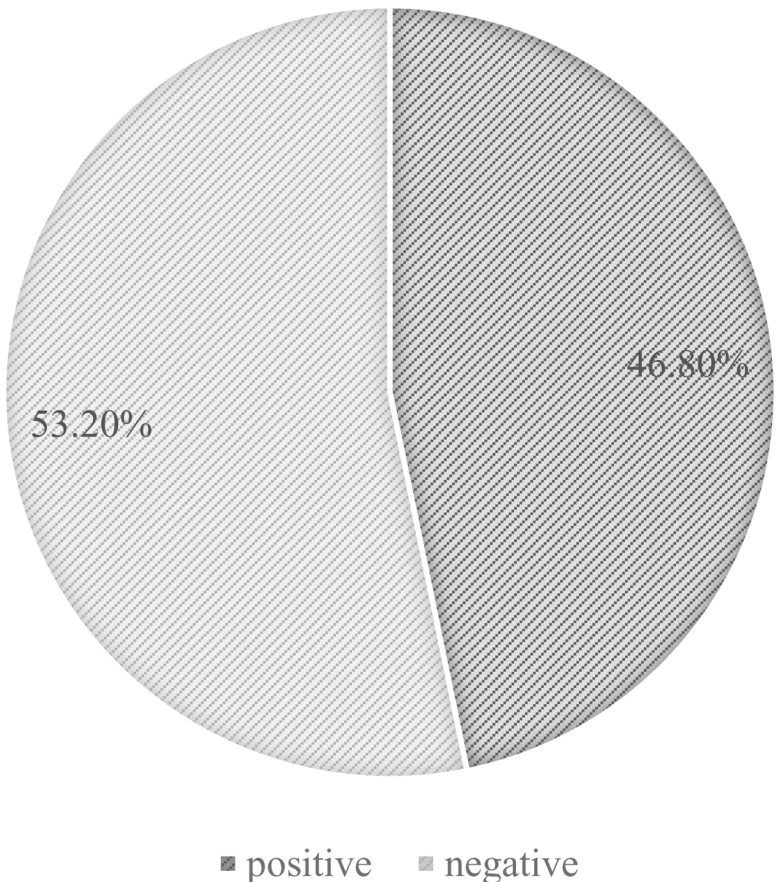

**Fig 2. Prevalence of malaria with microscopy.**

**Table 3. Prevalence of malaria symptoms and malaria with malaria RDT and microscopy.**

| Variable | Category | No (%) | 95%CI |
|---|---|---|---|
| **Presence of symptoms** | No | 107 (42.8) | 32-51.7 |
| | Yes | 143 (57.2) | 51.1-63.3 |
| | **Total** | **250 (100)** | |
| **RDT** | Negative | 190 (76) | 70.7-81.3 |
| | Positive | 60 (24) | 18.7-29.3 |
| | **Total** | **250 (100)** | |
| **Microscopy** | Negative | 117 (46.8) | 40.6-53.0 |
| | Positive | 133 (53.2) | 47.0-59.4 |
| | **Total** | **250 (100)** | |

the positive with microscopy turned out to be negative with mRDT (FN = 73). Sixty (60) samples were then Positive with RDT (TP = 60) and all of the 60 positives samples that were positives with RDT, were also positive with microscopy (FP = 0). Out of the one hundred and ninety (190) samples that were tested negative with RDT, one hundred and seventeen (117) were also negative with microscopy. The mRDT correctly identified 60 (TP) malaria cases out of 133 that were identified with microscopy and missed 73 (FN) malaria cases (Table 4).

The sensitivity and specificity were 45% (95% CI:37–54) and 100% (95%CI:96–100), respectively while the Positive Predictive Value (PPV), Negative Predictive Value (NPV) and accuracy were 100% (95% CI:94–100), 61.50% (95% CI:54–68) and 70.8% (95% CI:65–76) respectively for the malaria mRDT (Table 5). Cohen's Kappa analysis revealed a moderate degree of agreement between mRDT and microscopy (K = 0.435, p<0.001) (Table 5).

### Association between malaria parasites density and RDT positivity test

There was a statistically significant positive and moderate correlation between parasite density for malaria and malaria RDT (r = 0.459, p < 0.001) (Tables 6, 7).

## Discussion

In order to control malaria, a high-quality diagnostic technique must be used to identify the parasite before anti-malarial medication is prescribed in accordance with WHO guidelines. Diagnosing malaria parasites allows for early parasite identification, supports characterization of the treatment response, and focuses treatment efforts [22].

**Table 4. Validity of SD BIOLINE to diagnose *Plasmodium falciparum* and concurrence with Microscopy.**

| mRDT | | Microscopy | | Total |
|---|---|---|---|---|
| | | Positive | Negative | |
| | Positive | 60 | 0 | 60 |
| | Negative | 73 | 117 | 190 |
| | Total | 133 | 117 | 250 |

FN = False Negative; TP= True Positive; FP= False Positive.

**Table 5. Performance of mRDT in diagnosis of *P. falciparum*.**

| Measure | RDT% (95%CI) |
|---|---|
| Sensitivity | 45 (37-54) |
| Specificity | 100(96-100) |
| PPV | 100(94-100) |
| NPV | 61.50(54-68) |
| Kappa value | 0.435(0.369 - 0.435) |
| Accuracy | 70.8(65-76) |
| AUC value | 0.78 |

**Table 6. Correlation between Malaria parasites density and RDT positivity.**

| | | Parasite density | RDT |
|---|---|---|---|
| **Microscopy (Trophozoites/mm³)** | Pearson correlation | 1 | 0.459 |
| | P-value | | 0.000 |
| | N | 250 | 250 |
| **RDT** | Pearson Correlation | 0.459 | 1 |
| | P-value | 0.000 | |
| | N | 250 | 250 |

Table 7.  Association between Malaria parasites density and RDT positivity.

| Parasite density | RDT | | Chi-square | P-value |
|---|---|---|---|---|
| | Positive (%) | Total (%) | | |
| Low | 52 (20.8) | 121 (48.4) | 72.809 | <0.001 |
| Intermediate | 8 (3.2) | 12 (4.8) | | |
| High | 0 (0.0) | 117 (46.8) | | |
| Total | 60 (24) | 250 (100) | | |
| | 24.0% | 100.0% | | |

The prevalence of malaria with the mRDT was found to be 24% and 53.2% with microscopy in Tombel area with statistically significant difference (p < 0.05). This disparity could be attributed to some of the inherent and external factors, like the stability of the RDT which include exposure to extreme temperatures, which are a major contributor to poor performance of RDT, especially during transport as well as storage as observed by previous research. The prevalence with mRDT is lower than that recorded by Tchinda et al [23] in a study conducted in the Far North region of Cameroon with the overall malaria prevalence of 34.9% using mRDT and 59.2% using microscopy. In a study in the North West region of Cameroon, Kimbi and collaborators reported a malaria prevalence of 26.5% using mRDT which is similar to our study and 37.5% using microscopy which is lower than that recorded in our study [24]. These differences in prevalence may be due to the difference in endemicity of malaria from the different malaria epidemiological zones where the studies were carried.

Malaria prevalence was higher in females than males by microscopy and the findings are in line with Kimbi and collaborators [25] who reported that females spend more time outdoors at dusk and dawn than males to perform household chores and as such are more exposed to mosquito bites. Those aged 31-45 had a higher prevalence compared to the other age ranges which is contrary to the findings of Ning et al where children <5 years of age were more infected [26]. This could be explained by the fact that since it was a rural setting, the older adults are those involved with farming activities and businesses that keeps them outdoors exposes them to mosquito bites.

There was a statistically significant strong positive association between the presence of malaria symptoms and the mRDT positivity of (P= 0.023). In other words, individuals who reported malaria symptoms were more likely to test positive for malaria compared to those who did not have symptoms (OR = 2.056 times [95%CI; 1.10-3.82]). This is similar to a study carried out by Deus and collaborators 2019 in Tanzania who reported a statistically significant strong positive association between presence of malaria symptoms and RDT positivity.

The sensitivity and specificity of the RDT were 45% and 100% respectively. A sensitivity of 45% means that the test correctly identified 60 (45%) of true positive *P. falciparium* cases while specificity of 100% implies that the test correctly identified all negative cases. Also, the high PPV of 100% suggests that if the mRDT identifies a participant as positive for malaria, there is a very high likelihood that they are truly infected. However, the lower NPV of 61.5% suggests that if the test identifies a person as negative for malaria, there is still a chance that they might be infected. The lower sensitivity may be due to the quality of the RDT used and the prevalence of malaria in our study population. The findings of our study contradict the findings of Kimbi and collaborators [24] who reported a sensitivity and specificity of 82.4% and 76.6% respectively. Their study also reported a PPV of 57.4% and NPV of 91.9% which is different with our findings. Another study conducted in the South West region of Cameroon by Wanji and collaborators [25] reported findings similar to our study with a specificity being 98.3% and slightly different PPV of 95.9% but higher sensitivity of 91.2% and NPV of 97.3%.

These differences may be due to the difference in study population and difference in prevalence of the disease. A study by Maltha and collaborators [26] in Burkina Faso demonstrated higher sensitivity, specificity, PPV and NPV of 100%, 70.9%, 69.4% and 100%, respectively, for PfHRP2 detection. Likewise, a study by Diongue *et al.* [27] in Senegal reported that Carestat RDT showed high sensitivity (97.3%) and specificity (94.1%) with PPV and NPV of 97.3 and 94.1% respectively. Several factors are reported to affect the sensitivity of RDTs based on detection of HRP2, including an inherent limitation of the device, mutation or deletion of the gene encoding the HRP2, and storage conditions. It is known that the sensitivity of the RDT is affected by low parasitic densities, and that below 100 parasites/μL the mRDT performance decrease.

The area under the curve (AUC) was 0.78 for the receiver operating characteristic (ROC) curve of the malaria rapid diagnostic test (RDT) and this value suggests that the test has moderate diagnostic accuracy. The AUC of 0.78 means that the test has a 78% chance of correctly distinguishing between individuals who have malaria and those who do not have malaria. However, it also means that there is a 22% chance that the test will produce a false positive or false negative result. This level of accuracy may be acceptable in some contexts, but it is not ideal for a diagnostic test, particularly for a disease like malaria that can have serious health consequences if left untreated. The findings of this study are similar to a study carried out by Teh and collaborators [28] in the Mount Cameroon area who reported an AUC of 0.75. This could be due to the similarity in the study population and prevalence of the disease. On the other hand, our finding is lower than that of a study carried out by Njunkeng and collaborators [29] in Cameroon who reported an AUC of 0.97. This disparity might be due to the difference in population type and prevalence. Another study carried out by Mbulli and collaborators [30] in Cameroon demonstrated a higher AUC of 0.90. This level of accuracy which is higher than that of our study may be associated to the fact that their study were children <5 years who were 2 times more likely to be positive for mRDT than the older.

A significant association (*P*<0.001) was identified between the SD BIOLINE malaria Ag *P. f* RDT and the parasite density. This is similar to a study carried out by Teh and collaborators [28] in Cameroon which had a sensitivity of 96.1% at parasite density greater than or equal to 200 parasites/μL of blood, as recommended by WHO [31]. This is similar to the findings of Oyibo et al [32] in a study who identified a significant association between parasite density and RDT positivity in children aged 0-2 years in Nigeria, with higher parasite densities associated with greater likelihood of RDT positivity. Lopes et al [33] reported a similar finding for which there was a significant association between parasite density and RDT positivity for the SD Bioline Malaria Ag *P.f.* Another study carried out by Djoufounna et al. [34] found a significant association between parasite density and RDT positivity for the CareStart Malaria HRP2 RDT. This implies an increasing parasite density increased the sensitivity of the mRDT to detect the presence of the malaria parasite.

The test utilized for this investigation was based on the HRP2, which is the most widely used test for parasitological confirmation of malaria before treatment in sub-Saharan Africa. Numerous studies have reported marked drops in HRP2 RDT sensitivity following reductions in P.f HRP2 deletion or transmission intensity. Parasites missing HRP2 are a potential source of false-negative HRP2 RDTs. Although RDTs are employed as diagnostic tools, diagnosis by microscopy should never be abandoned because it is the gold standard in endemic areas. In addition, microscopy allows the computation of parasite densities and identification of all species and is cheaper than the other approaches. PCR is still expensive even though it is the most effective diagnostic technique with excellent sensitivity and specificity.

## Conclusions

The overall prevalence of malaria is high in the Tombel Health Area both with use of the mRDT and microscopy. SD Bioline Malaria Ag P.f used reported a high level of specificity and positive predictive value with relatively low sensitivity and negative predictive value. In addition, the overall accuracy of SD BIOLINE Malaria Antigen P.f HRP2 with microscopy as reference standard in population screening for *P. falciparum* infection in the community is moderate. The RDT has proven to be moderately effective as a diagnostic test for confirmation of clinical cases of malaria. This study shows a variation in the performance of the RDT with respect to the age. Hence this sociodemographic characteristic must be taken into account, in order to explain the results obtained. This, in a certain way, indicates that the use of RDTs is limited, it may be useful for diagnoses in remote areas, but for prevalence studies it is better to continue using microscopy as a reference technique or if it is possible the PCR. Finally, there is a significant association between parasite density and the RDT positivity where an increase parasite density increases the sensitivity of the RDT ($P<0.001$). Microscopy remains a preferable option, when parasite density needs to be determined in the absence of PCR.

## Supporting information

**S1 Data. Diagnostic test calculation.**
(PDF)

**S2 Data. Data research.**
(XLSX)

**S3 Data. Data analysis.**
(XLSX)

**S4 Data. Diagnostic test computed.**
(PDF)

**S5 Data. SPSS computed.**
(PDF)

**S6 Data. Malaria laboratory protocol.**
(DOCX)

## Author contributions

**Conceptualization:** Abigail L. Ngalame, Dickson S. Nsagha.

**Data curation:** Abigail L. Ngalame, Djakissam Watching, Elisabeth M. Zeukóo, Dickson S. Nsagha.

**Formal analysis:** Abigail L. Ngalame, Djakissam Watching, Odette Z. Kibu.

**Investigation:** Abigail L. Ngalame.

**Methodology:** Abigail L. Ngalame, Djakissam Watching, Dickson S. Nsagha.

**Project administration:** Dickson S. Nsagha.

**Supervision:** Djakissam Watching, Dickson S. Nsagha.

**Validation:** Djakissam Watching, Odette Z. Kibu, Elisabeth M. Zeukóo, Dickson S. Nsagha.

**Writing – original draft:** Abigail L. Ngalame, Djakissam Watching.

**Writing – review & editing:** Odette Z. Kibu, Elisabeth M. Zeukóo, Dickson S. Nsagha.

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
