## [Decision Letter · Decision Letter 0]

2 May 2024

PONE-D-23-43663Assessment of the diagnostic performance of the SD Bioline malaria antigen Plasmodium falciparum rapid diagnostic test for the diagnosis of malaria in the Tombel health area, Southwest region of CameroonPLOS ONE

Dear Dr. Djakissam,

Thank you for submitting your manuscript to PLOS ONE. After careful consideration, we feel that it has merit but does not fully meet PLOS ONE’s publication criteria as it currently stands. Therefore, we invite you to submit a revised version of the manuscript that addresses the points raised during the review process.

We look forward to receiving your revised manuscript.

Kind regards,

Enoch Aninagyei, PhD

Academic Editor

PLOS ONE

2. PLOS requires an ORCID iD for the corresponding author in Editorial Manager on papers submitted after December 6th, 2016. Please ensure that you have an ORCID iD and that it is validated in Editorial Manager. To do this, go to ‘Update my Information’ (in the upper left-hand corner of the main menu), and click on the Fetch/Validate link next to the ORCID field. This will take you to the ORCID site and allow you to create a new iD or authenticate a pre-existing iD in Editorial Manager. Please see the following video for instructions on linking an ORCID iD to your Editorial Manager account: https://www.youtube.com/watch?v=_xcclfuvtxQ".

Reviewers' comments:

Reviewer's Responses to Questions

**Comments to the Author**

1. Is the manuscript technically sound, and do the data support the conclusions?

Reviewer #1: No

Reviewer #2: No

Reviewer #3: Partly

Reviewer #4: Partly

Reviewer #5: Yes

Reviewer #6: Yes

Reviewer #7: Yes

2. Has the statistical analysis been performed appropriately and rigorously? 

Reviewer #1: No

Reviewer #2: No

Reviewer #3: Yes

Reviewer #4: No

Reviewer #5: Yes

Reviewer #6: Yes

Reviewer #7: Yes

3. Have the authors made all data underlying the findings in their manuscript fully available?

Reviewer #1: No

Reviewer #2: Yes

Reviewer #3: Yes

Reviewer #4: No

Reviewer #5: Yes

Reviewer #6: Yes

Reviewer #7: Yes

4. Is the manuscript presented in an intelligible fashion and written in standard English?

Reviewer #1: Yes

Reviewer #2: No

Reviewer #3: Yes

Reviewer #4: No

Reviewer #5: Yes

Reviewer #6: Yes

Reviewer #7: Yes

5. Review Comments to the Author

Reviewer #1: The manuscript by Ngalame et al. evaluates the performance of the SD Bioline malaria antigen Plasmodium falciparum rapid diagnostic test (RDT) compared to microscopy for malaria diagnosis in a cross-sectional community-based study in the Tombel Health Area, Cameroon. The study involved 250 participants and found a malaria prevalence of 24% using the RDT and 53.2% using microscopy. The RDT demonstrated a sensitivity of 45%, specificity of 100%, positive predictive value of 100%, and negative predictive value of 61.5%. The area under the ROC curve was 0.78, indicating moderate diagnostic accuracy. The authors conclude that the overall accuracy of the RDT is moderate and may not be ideal for a disease like malaria with serious health consequences if left untreated. While the study addresses an important topic and provides valuable data on the performance of the RDT in a rural setting, there are several concerns regarding the methodology, data analysis, and interpretation that need to be addressed. The manuscript requires substantial revisions to improve its overall quality. However, I have major critiques/concerns/comments/suggestions and minor points to be considered:

Major

a. The sample size calculation is unclear. The authors mention using the Burderer method but do not provide sufficient details on the assumptions and values used in the calculation.

b. The authors used a consecutive sampling technique based on the quarters in the Tombel Health Area. However, they do not provide a clear rationale for this approach or discuss potential limitations, such as selection bias.

c. The authors do not mention blinding of the microscopists to the RDT results. This is important to avoid bias in the interpretation of the microscopy results.

d. The data analysis section lacks detail. The authors should provide more information on how they handled missing data, if any, and how they calculated the 95% confidence intervals for the performance measures.

e. The authors should provide a more in-depth comparison of their findings with other studies and discuss potential reasons for the differences observed. They should also discuss the implications of their findings for malaria diagnosis and control in the study area and similar settings.

Minor

f. The authors should provide more information on the training and experience of the microscopists who interpreted the blood films.

g. The proofreading is mandatory!

h. The authors should ensure that all references are formatted consistently and following the journal's guidelines.

Reviewer #2: The study does not appear to be sound and the flow of the manuscript was inconsistent and did not meet the PLOS ONE publication criteria. In particular, the order of the scientific flow was not maintained in the Methods and Materials section, and there was controversial information in the manuscript.

For instance:

1. The study period in the Abstract was “March to May 2023" and in the Materials and Methods section "February to July 2023".

2. The total population of the study area was not clear with large differences; the author has given an approximate total (65,000 people) and the sum of the numbers of males, females and children 0-16 years (89,178 people). See page 13 Study area and setting section paragraph 1 “The THD is situated in a rural area, with a population of approximately 65000 people. The major sociological groups are Bakossi, Grass landers, Nigerians, Banyangi, Oroko. In general, the population consists of-Approximately 35,690 males Approximately 28,587 females– And about 24 901 children aged 0-16 years”.

3. The author stated in the Methods and Materials section that the study population was > 5 years and above, and in the Results section that the author included > 2 years of age. Also, the author does not explain in the inclusion and exclusion criteria why only age > 5 years and above and not under 5 years.

4. The sample size mentioned by the author in the abstract was 250, in the sample size determination 239, in the proportional allocation in table1 260, and in the result section 250, so which was the correct sample size??

5. The author also used a consecutive sampling technique, why consecutive? Because it leads to bias or not representative to the population in the area, one of the reasons to use consecutive sampling technique was limited case or study population, but in this study the study was community based, so there was no challenge to get the determined sample size in the aforementioned study period. The way the sample was allocated was also not clear.

The technical details were also unclear and had no scientific basis. For instance; data collection and laboratory diagnosis, the author does not follow the guidelines or SOPs of the given laboratory at all during the procedures and examination.

1. Who was the data collector?

2. Who carried out the laboratory investigation?

3. What was the scientific basis regarding to the way of thick and thin film preparation?

4. ‘Small amount of blood’, how much was small?

5. Using an applicator stick to prepare the smear and lyse RBCs "until complete hemolysis of the RBCs" What was your scientific evidence?

6. Data management and analysis was mixed with the laboratory examination, the parasite density calculation was to be included under microscopic examination but not the data analysis.

7. The statistical analysis was not carried out appropriately and rigorously; on page number ‘20” the author stated that the formula for calculating PPV and NPV was incorrect.

In order to review the results and discussion section, the comments in the methods and materials section should be addressed first.

With kind regards!

Reviewer #3: The manuscript was conducted with full ethical standards and received ethical clearance, as well as administrative authorizations and an informed consent from participants. The weaknesses of the study include a limited range because the research was focused only on the diagnostic performance of the one type RDT in the one particular region, so can hamper its generalization in other settings. Noted sample size , while the sample calculation was included in the methodology chapter, the final 250 individuals might be insufficient for the appropriate approximation.

Reviewer #4: Dear Authors,

Thank you for your manuscript, here are some comments about the work:

1- Please add some specificity in the writing "Malaria continues to be a serious global public health concern", I think its is a problem in certain regions of the world and not globally.

2- please revise the English language

3- The sampling process is general and not specific how the samples were conducted "Study population

All persons living in the Tombel Health Area aged from 5 years and above" I Think this is very general, please rewrite and specify

4- There is no statistical data or analysis to the data, please specify and mention the details

5- There is no need to repeat the results data once as data in the text and in tables, please avoid redundancy

Reviewer #5: Comment

- Title: too long correct it as “Assessment of the diagnostic performance of the SD Bioline malaria

antigen test for the diagnosis of malaria in the Tombel health area, Southwest region of Cameroon”

- abstract: conclusion not clear

- Introduction: grammar problem

- Method: study design not clear, is it institutional / community based?

- Study Period: data collection period not clear, there is difference between the body of the manuscript and abstract.

- Have you done pretest for questionnaire?

- In general the manuscript needs major revision

The author should revise the manuscript critically, unless it is nearer to rejection.

Reviewer #6: Comments on Topics: - ‘’Assessment of the diagnostic performance of the SD Bioline malaria antigen Plasmodium falciparum rapid diagnostic test for the diagnosis of malaria in the Tombel health area, Southwest region of Cameroon’’.

1. Abstract Section

22% chance that the test can produce a false result-this is there is high chance of false positive or false negative.

AUC…what it means?

Low sensitivity and Overall accuracy of P.f HRP2 in the THA is moderate so how you recommend for diagnosis of malaria.

2. Materials and Methods

Study Period -Different study period was noted (March to May 2023, February to July 2023,30th of April 2023 and was ended on the 20th of June 2023).Which is correct from this all?

Very large statement about study area

Exclusion and inclusion criteria: redundant words.

Sampling technics change into Sampling technique.

Consecutive sampling vs Probability proportionate to size –these both are quite different sampling techniques so how do you use simultaneously?

Data management and analysis- what types of Logistic regression analysis?

3. Results

Age ranged from 2-70 years vs study population aged from 5 years and above?

Table 2: -Highest Level what it mean?

-Why you include Religion or is there is any relationship with your topics? Since this is sensitive case it’s better to omit religion.

Generally is the manuscript is too long thus it seems like the original paper rather than manuscript. So please the check the PLOS ONE guidelines and work accordingly.

Reviewer #7: The research was thoughtful done with you following research ethics and publication ethics. The statistical analysis carefully done appropriately and rigorously. Though I would appreciate that the Research is extended to parts of Cameroon

6. PLOS authors have the option to publish the peer review history of their article (what does this mean? ). If published, this will include your full peer review and any attached files.

**Do you want your identity to be public for this peer review?** For information about this choice, including consent withdrawal, please see our Privacy Policy .

Reviewer #1: No

Reviewer #2: No

Reviewer #3: No

Reviewer #4: No

Reviewer #5: No

Reviewer #6: **Yes: ** Fedasan Alemu Abdi

Reviewer #7: No

---

## [Author Response · Author response to Decision Letter 1]

27 Jun 2024

PONE-D-23-43663

Response to Reviewers

June 16th, 2024

Dear Dr Enoch,

Thank you for giving us the opportunity to submit a revised draft of the manuscript entitled “Assessment of the diagnostic performance of the SD Bioline malaria antigen Plasmodium falciparum rapid diagnostic test for the diagnosis of malaria in the Tombel health area, Southwest region of Cameroon” for publication in the PLOS ONE. We appreciate the time and effort that you and the reviewers dedicated to providing feedback on our manuscript and are grateful for the insightful comments on and valuable improvements to our paper. We have incorporated most of the suggestions made by the reviewers. Those changes are highlighted within the manuscript. Please see below, in blue, for a point-by-point response to the reviewers’ comments and concerns.

Reviewers' Comments to the Authors:

Reviewer 1

a. Comment from Reviewer 1: The sample size calculation is unclear. The authors mention using the Burderer method but do not provide sufficient details on the assumptions and values used in the calculation.

Authors’ response: Thank you for pointing this out. The reviewer is correct,

We used the sample size calculation method of Burder and collaborators 1996 as follow:

n = n1 + n2

And

Where,

n1 and n2 = minimum required number of participants for sensitivity and specificity with estimated sensitivity and specificity of 95% and 99% respectively

TP = True Positive (Positive for malaria with microscopy)

TN = True Negative (Negative for malaria with microscopy)

FN = False Negative (Positive for malaria with microscopy but negative with the RDT)

FP = False Positive (Negative for malaria with microscopy but positive with the RDT)

P = prevalence of malaria from previous study (34.7% gotten from a study carried out by Ngum and collaborators., 2023 in Mezam division, North-West Region, Cameroon) [73].

Z = standard normal variation at a confidence level of 95% (standard value of 1.96)

W = 0.05 (5% maximum acceptable width of the 95% confidence interval)

Substituting in the equation above,

TP+FN=〖1.96〗^2 (0.95(1-0.95))/〖0.05〗^2

TP + FN = 72.99

TN+FP=〖1.96〗^2 (0.99(1-0.99))/〖0.05〗^2

TN + FP = 15.21

n1=72.99/0.347

n1 = 210.3

n2=15.21/ (1-0.347)

n2 = 23.3

n1 required for sensitivity = 211 participants n2 required for specificity = 24 participants (at least n = 235 participants).

Taking into account a 10 % non-response rate, our minimum sample size will be 258.5 ≈ 260 participants.

Due to the fact that we had a shorter period for data collection, we were unable to get more than 250 samples.

We have provided the detail in the manuscript.

b. Comment from Reviewer 1: The authors used a consecutive sampling technique based on the quarters in the Tombel Health Area. However, they do not provide a clear rationale for this approach or discuss potential limitations, such as selection bias.

Authors’ response: Thank you for pointing this out. This is true, but out of the 11 identified quarters, we randomly selected 9 which is representative of the population and we used probability proportionate to size to get the sample size per quarter. Updated in the manuscript

c. Comment from Reviewer 1: The authors do not mention blinding of the microscopists to the RDT results. This is important to avoid bias in the interpretation of the microscopy results.

Authors’ response: Thank you for pointing this out. That was not mentioned, was an error, but the microscopist was blinded of the RDT results, gotten in the field by the principal investigator and the thick and thin film slides were read in the Tombel health district by a trained microscopist.

Added in the manuscript

d. Comment from Reviewer 1: The data analysis section lacks detail. The authors should provide more information on how they handled missing data, if any, and how they calculated the 95% confidence intervals for the performance measures.

Authors’ response: Thank you for pointing this out. Data was keyed in by the principal investigator and this was done immediately after collection same day eliminating room for missing data. Updated in the manuscript

e. Comment from Reviewer 1: The authors should provide a more in-depth comparison of their findings with other studies and discuss potential reasons for the differences observed. They should also discuss the implications of their findings for malaria diagnosis and control in the study area and similar settings.

Author response: We agree with the reviewer’s assessment. Accordingly, throughout the manuscript, we have revised

f. Comment from Reviewer 1: The authors should provide more information on the training and experience of the microscopists who interpreted the blood films.

Author response: Thank you for pointing this out. The microscopist is a trained and professional microscopist working for the Tombel district Hospital. It has been added in the manuscript

g. Comment from Reviewer 1: The proofreading is mandatory

Author response: The entire manuscript has been proofread.

h. Comment from Reviewer 1: The authors should ensure that all references are formatted consistently and following the journal's guidelines.

Author response: Thank you for pointing this out. The reviewer is correct, and the references are updated

Reviewer #2:

The study does not appear to be sound and the flow of the manuscript was inconsistent and did not meet the PLOS ONE publication criteria. In particular, the order of the scientific flow was not maintained in the Methods and Materials section, and there was controversial information in the manuscript.

Author response: While we appreciate the reviewer’s feedback, we respectfully disagree. We think this study makes a valuable contribution to the field because the study provides valuable information on the accuracy of the RDT used in the THA and its potential usefulness in improving the diagnosis and treatment of malaria in rural areas for the PLOS ONE’s readers. The findings from this study will inform policy makers, healthcare providers, and other stakeholders involved in the management of malaria, and ultimately contribute to the reduction of the burden of this disease in rural communities.

1. Comment from Reviewer 2: The study period in the Abstract was “March to May 2023" and in the Materials and Methods section "February to July 2023".

Author response: Thank you for pointing this out. The reviewer is correct, that was a typographical error. Study period was from February to July 2023 comprising protocol development, collection of data, data management and analysis. The period of April to June was only for data collection, corrected in the manuscript

2. Comment from Reviewer 2: The total population of the study area was not clear with large differences; the author has given an approximate total (65,000 people) and the sum of the numbers of males, females and children 0-16 years (89,178 people). See page 13 Study area and setting section paragraph 1 “The THD is situated in a rural area, with a population of approximately 65000 people. The major sociological groups are Bakossi, Grass landers, Nigerians, Banyangi, Oroko. In general, the population consists of-Approximately 35,690 males Approximately 28,587 females– And about 24 901 children aged 0-16 years”.

Author response: Thank you for this suggestion. The population size was a representation of males and females with children reason we had approximately 65,000 thousand people (35,690 males + 28,587 females = 64227 which is approximately 65000). “With” replaced by “and” in the manuscript

3. Comment from Reviewer 2: The author stated in the Methods and Materials section that the study population was > 5 years and above, and in the Results section that the author included > 2 years of age. Also, the author does not explain in the inclusion and exclusion criteria why only age > 5 years and above and not under 5 years

Author response: Thank you for pointing this out. The reviewer is correct, it was a typographical error, participants were all age. Corrected in the manuscript

4. Comment from Reviewer 2: The sample size mentioned by the author in the abstract was 250, in the sample size determination 239, in the proportional allocation in table1 260, and in the result section 250, so which was the correct sample size??

Author response: Thank you for pointing this out. The details for this section have been provided in the manuscript and has been explained (see author’s response to comment a. from reviewer 1).

5. Comment from Reviewer 2: The author also used a consecutive sampling technique, why consecutive? Because it leads to bias or not representative to the population in the area, one of the reasons to use consecutive sampling technique was limited case or study population, but in this study the study was community based, so there was no challenge to get the determined sample size in the aforementioned study period. The way the sample was allocated was also not clear.

Authors’ response: This is true, but out of the 11 identified quarters, we randomly selected 9 which is representative of the population and we used probability proportionate to size to get the sample size per quarter.

Comment from Reviewer 2: The technical details were also unclear and had no scientific basis. For instance; data collection and laboratory diagnosis, the author does not follow the guidelines or SOPs of the given laboratory at all during the procedures and examination.

1. Who was the data collector?

Authors’ response: Thank you for pointing this out. Principal investigator who is also a certified and experienced laboratory scientist; Ngalame Abigail. Was added in the manuscript

2. Who carried out the laboratory investigation?

Authors’ response: Thank you for pointing this out. A trained microscopist employed at the Tombel District Hospital: Mr. Macleo. Was added in the manuscript

3. Comment from Reviewer 2: What was the scientific basis regarding to the way of thick and thin film preparation?

Authors’ response:

Thick film;

1. The slides were labelled with the participant's unique identification number

2. Approximately a drop of 6 µL of blood was placed in the centre the slide.

3. The applicator stick was used to spread the drop in a circular pattern until the RBCs are completely hemolyzed and it has the size of a dime.

4. The slides were laid on a smooth surface and allowed to air dry thoroughly at room temperature.

5. Stained with giemsa, rinsed and let to air dry.

Thin film;

1. Approximately a drop of 3 µL of blood was placed on a clean labelled slide near its frosted end.

2. Another slide was placed at a 45-degree angle up to the drop, allowing the drop to spread along the contact line of the two slides and a thin layer of even spread was made.

3. The thin smears were allowed to dry and later fixed by immersing them into absolute methanol for a minute.

4. Stained with Giemsa, rinsed and let to air dry.

5. Properly hemolysed RBCs retain the stain which was obvious after rinsing the stained slides

6. Comment from Reviewer 2: Data management and analysis was mixed with the laboratory examination; the parasite density calculation was to be included under microscopic examination but not the data analysis.

Author response: Thank you for pointing this out. Data management and analysis and the laboratory examination sections have been corrected

7. Comment from Reviewer 2: The statistical analysis was not carried out appropriately and rigorously; on page number ‘20” the author stated that the formula for calculating PPV and NPV was incorrect.

Author response: Thank you for pointing this out. The statistical analysis section has been updated

Reviewer #3:

Comment from Reviewer 3: The manuscript was conducted with full ethical standards and received ethical clearance, as well as administrative authorizations and an informed consent from participants. The weaknesses of the study include a limited range because the research was focused only on the diagnostic performance of the one type RDT in the one particular region, so can hamper its generalization in other settings. Noted sample size, while the sample calculation was included in the methodology chapter, the final 250 individuals might be insufficient for the appropriate approximation.

Authors’ response: Thank you for this suggestion. It would have been interesting to extend this study to other localities. However, this would not be possible because the focus was on the particular RDT constantly used in the Tombel District Hospital for testing malaria. Also, due to the fact that we had a shorter period for data collection, we were unable to get more than 250 samples. The justification has been added

Reviewer #4:

1- Comment from Reviewer 4: Please add some specificity in the writing "Malaria continues to be a serious global public health concern", I think it is a problem in certain regions of the world and not globally.

Authors’ response: As suggested by the reviewer, The revised text reads as follows Malaria continues to be a serious global public health concern. The World Health Organization (WHO) reported 405,000 malaria deaths and 228 million cases in 2018 worldwide in all endemic areas.

2- Comment from Reviewer 4: please revise the English language

Authors’ response: Thank you for this remark. English language has been revised in general.

3- Comment from Reviewer 4: The sampling process is general and not specific how the samples were conducted "Study population

Authors’ response: reviewer 1 addressed the same issue in comment a. (has been answered)

Comment from Reviewer 4: All persons living in the Tombel Health Area aged from 5 years and above" I Think this is very general, please rewrite and specify

All persons living in the Tombel Health Area during recruitment of participants for the study aged from 2 years and above.

Authors’ response: reviewer 2 addressed the same issue in comment 3. (has been answered)

4- Comment from Reviewer 4: There is no statistical data or analysis to the data, please specify and mention the details

Authors’ response: reviewer 1 addressed the same issue in comment d. (updated in the manuscript)

5- Comment from Reviewer 4: There is no need to repeat the results data once as data in the text and in tables, please avoid redundancy

Authors’ response: Thank you for this remark. Text has been revised in general.

Reviewer #5:

Authors’ response: Thank you for all remarks. Those issues have been revised in the manuscript as indicated

- Comment from Reviewer 5: Title: too long correct it as “Assessment of the diagnostic performance of the SD Bioline malaria antigen test for the diagnosis of malaria in the Tombel health area, Southwest region of Cameroon”

- abstract: conclusion not clear

- Introduction: grammar problem

- Method: study design not clear, is it institutional / community based?

- Study Period: data collection period not clear, there is difference between the body of the manuscript and abstract.

- Have you done pretest for questionnaire?

Reviewer #6:

1. Abstract Section

- 22% chance that the test can produce a false result-this is there is high chance of false positive or false negative.

Authors’ response: The AUC of 0.78 means that the test has a 78% chance of correctly distinguishing between individuals who have malaria and those who do not have malaria. However, it also means that there is a 22% chance that the test will produce a false positive or false negative result.

- AUC…what it means?

Authors’ response: AUC = Area Under the (Receiver Operating curve; ROC) Curve

- Low sensitivity and Overall accuracy of P.f HRP2 in the THA is moderate so how you recommend for diagnosis of malaria.

Authors’ response: This level of accuracy may be acceptable in some contexts, but it is not ideal for a diagnostic screening test, particularly for a disease like malaria that can have serious health consequences if left untreated.

2. Materials and Methods

- Study Period -Different study period was noted (March to May 2023, February to July 2023,30th of April 2023 and was ended on the 20th of June 2023). Which is correct from this all?

Author response: Thank you for pointing this out. The revie

---

## [Decision Letter · Decision Letter 1]

1 Aug 2024

PONE-D-23-43663R1Assessment of the diagnostic performance of the SD Bioline malaria antigen test for the diagnosis of malaria in the Tombel health area, Southwest region of CameroonPLOS ONE

Dear Dr. Djakissam,

Thank you for submitting your manuscript to PLOS ONE. After careful consideration, we feel that it has merit but does not fully meet PLOS ONE’s publication criteria as it currently stands. Therefore, we invite you to submit a revised version of the manuscript that addresses the points raised during the review process.

**ACADEMIC EDITOR:**

Abstract

The abstract should be rewritten in an unstructured formatRevise ‘The aim of this study was to assess the diagnostic performance Ag P.f RDT…’ to read ‘The aim of this study was to assess the diagnostic performance of the the SD Bioline Malaria Ag P.f RDT…’A consecutive sampling technique was used to collect blood samples of 250 individuals…were they asymptomatic individuals? Please clarifiyMicroscopy identified 133 (53.2%) participants with malaria…clinical or asymptomatic?Microscopy identified 133 (53.2%) participants..include the parasite density rangeDefine mRDT on first mention… mRDT identified 60 cases add % and include the parasite density range detected by the mRDT

Introduction

405,000 malaria deaths and 228 million cases in 2018..provide current data. How different is this statement from the succeeding one? Merge if possibleProvide reference for ‘Compared to 625,000 in 2020, the expected number of malaria deaths in 2021 was 619 000. In Sub-Saharan Africa (SSA), the figures for cases and fatalities were 93% and 94% respectively’These include a lengthy return time, would you consider turnaround time?Authors use mRDT and in some places RDT, please be consistentDefine THA on first mention

Methods

Combine study design and study period to read ‘Study design and period’In the abstract you mentioned that sample were collected from March to May 2023 but under the method sections, you mentioned 30th of April 2023 to 20th of June 2023. Please clarify these differences in the study periodsWhat is the difference between THA and THD?The Study area and setting section is too lengthy. Be brief by removing all irrelevant and unrelated information.  Under inclusion/exclusion, write the information in continuous sentences.Sample size determination. Your method look quite confusing. Why don’t you use the cochrane formula once you know the prevalence of malaria from a previous study. If you are too strong about this method, show previous studies that used that methodChange Sampling technics to Sampling techniquesA drop of blood was placed in the centre the slide. State exact volumeUnder thick film, please revise the information. It is not the applicator stick that hemolyses the samples and this is not why the sample is spread on the slideUnder thin film, state the exact volume as wellUnder microscopy, check the spelling of specieProcedure for RDT Test: Approximately 5μL of blood were used to diagnose malaria using the malaria antigen Plasmodium falciparum RDT kit, rephrase this statement. The volume of the blood is used to perform the test not used to diagnose malaria. Describe how the test was done and not the princples behind the test

Results

Table 3: change NO to No. Same to YESDefine all the abbreviations under Validity of SD BIOLINE to diagnose P. falciparum and concurrence with MicroscopyTable 5: Performance of mRDT in diagnosis of P. falciparum compared to reference standard and studies in Cameroon. I did not see any previous study comparisonTable 7, how do you classify parasite count as low, high, moderate etc. Justify these classification with reference to a published paper

Discussion

The discussion section had your results repeated. Please discussion your findings in reference to malaria control in your country. Do not repeat the resultsBe consistent with the use of HRP2, the discussion section has HRPII

We look forward to receiving your revised manuscript.

Kind regards,

Enoch Aninagyei, PhD

Academic Editor

PLOS ONE

Journal Requirements:

Reviewers' comments:

Reviewer's Responses to Questions

**Comments to the Author**

1. If the authors have adequately addressed your comments raised in a previous round of review and you feel that this manuscript is now acceptable for publication, you may indicate that here to bypass the “Comments to the Author” section, enter your conflict of interest statement in the “Confidential to Editor” section, and submit your "Accept" recommendation.

Reviewer #4: All comments have been addressed

Reviewer #6: All comments have been addressed

2. Is the manuscript technically sound, and do the data support the conclusions?

Reviewer #4: Partly

Reviewer #6: Yes

3. Has the statistical analysis been performed appropriately and rigorously? 

Reviewer #4: Yes

Reviewer #6: Yes

4. Have the authors made all data underlying the findings in their manuscript fully available?

Reviewer #4: Yes

Reviewer #6: Yes

5. Is the manuscript presented in an intelligible fashion and written in standard English?

Reviewer #4: Yes

Reviewer #6: Yes

6. Review Comments to the Author

Reviewer #4: (No Response)

Reviewer #6: Thanks for your corrections. However some issues that I have raised was not touched till now.

Please check about the diagnosis sensitivity of SD Bioline malaria antigen test.

i.e. 22% chance that the test can produce a false result-this is there is high chance of false positive or false negative.

7. PLOS authors have the option to publish the peer review history of their article (what does this mean? ). If published, this will include your full peer review and any attached files.

**Do you want your identity to be public for this peer review?** For information about this choice, including consent withdrawal, please see our Privacy Policy .

Reviewer #4: **Yes: ** Walid Aburayyan

Reviewer #6: **Yes: ** Fedasan Alemu Abdi

---

## [Author Response · Author response to Decision Letter 2]

3 Sep 2024

Dear Dr. Enoch,

Thank you for giving me the opportunity to submit a revised draft of my manuscript titled [Assessment of the diagnostic performance of the SD Bioline malaria antigen test for the diagnosis of malaria in the Tombel health area, Southwest region of Cameroon] to [PLOS ONE]. We appreciate the time and effort that you and the reviewers have dedicated to providing your valuable feedback on our manuscript. We are grateful to the reviewers for their insightful comments on our paper. We have been able to incorporate changes to reflect most of the suggestions provided by the reviewers. We have highlighted the changes within the manuscript.

Here is a point-by-point response to the reviewers’ comments and concerns.

ABSTRACT

Comments from Reviewer 1

1 Comment 1: [The abstract should be rewritten in an unstructured format.]

Response: Thank you for pointing this out. We agree with this comment. Therefore, we have rewritten the abstract in an unstructured format

2 Comment 2: [Revise ‘The aim of this study was to assess the diagnostic performance Ag P.f RDT…’ to read ‘The aim of this study was to assess the diagnostic performance of the the SD Bioline Malaria Ag P.f RDT….]

Response: Agree. We have, revised this point.

3 Comment 3: [A consecutive sampling technique was used to collect blood samples of 250 individuals…were they asymptomatic individuals? Please clarify.]

Response: Thank you for pointing this out, the samples were collected from symptomatic and asymptomatic individuals. This have been incorporated throughout the manuscript.

4 Comment 4: [Microscopy identified 133 (53.2%) participants with malaria…clinical or asymptomatic?]

Response: Thank you for pointing this out, the microscopy identified positive cases in both symptomatic and asymptomatic individuals. This have been incorporated throughout the manuscript.

5 Comment 5: [Microscopy identified 133 (53.2%) participants include the parasite density range.]

Response: Thank you for this suggestion. According to the results in table 7 we obtain express the parasites densities range for mRDT

6 Comment 6: [Define mRDT on first mention.]

Response: Agree. We have, revised this point.

7 Comment 7: [mRDT identified 60 cases add % and include the parasite density range detected by the mRDT.]

Response: Thank you for this suggestion. 52 (20.8%) were low parasites density and 8 (3.2%) were moderate parasites density and 0.00 for high density. It has been added in the manuscript.

INTRODUCTION

1 Comment 1: [405,000 malaria deaths and 228 million cases in 2018.provide current data. How different is this statement from the succeeding one? Merge if possible.]

Response: thank you for this suggestion. In April 2024, the new World Malaria report has mentioned that, there were 608 000 malaria deaths in 2022 in globally. African continent was home to 94% of malaria cases (233 million) and 95% (580 000) of malaria deaths and Over 90% of all malaria deaths occur in Sub-Saharan Africa. [3]. Has been corrected within the manuscript.

2 Comment 2: [2. Provide reference for ‘Compared to 625,000 in 2020, the expected number of malaria deaths in 2021 was 619 000. In Sub-Saharan Africa (SSA), the figures for cases and fatalities were 93% and 94% respectively’.]

Response: thank you for this suggestion. This part has been replaced with In April 2024, the new World Malaria report has mentioned that, there were 608 000 malaria deaths in 2022 in globally. African continent was home to 94% of malaria cases (233 milli on) and 95% (580 000) of malaria deaths and Over 90% of all malaria deaths occur in Sub-Saharan Africa.

3 Comment 3: [These include a lengthy return time; would you consider turnaround time?]

Response: Thank you for pointing this out, we have, revised this point.

4 Comment 4: [Authors use mRDT and in some places RDT, please be consistent.]

Response: Thank you for pointing this out, we have, revised this point.

5 Comment 5: [Define THA on first mention.]

Response: Thank you for pointing this out, THA was replaced by Tombel Health District (THD) throughout the manuscript.

METHODS

6 Comment 6: [Combine study design and study period to read ‘Study design and period’.]

Response: Thank you for pointing this out, we have, revised this point.

6 Comment 6: [Combine study design and study period to read ‘Study design and period’.]

Response: Thank you for pointing this out, we have, revised this point.

7 Comment 7: [In the abstract you mentioned that sample were collected from March to May 2023 but under the method sections, you mentioned 30th of April 2023 to 20th of June 2023. Please clarify these differences in the study periods.]

Response: Thank you for pointing this out, sample were collected from 30th of April 2023 to 20th of June 2023. Has been corrected in the manuscript

8 Comment 8: [What is the difference between THA and THD?]

Response: Thank you for pointing this out, Tombel Health Area (THA) and Tombel Health District (THD) are the same area. THA has been replaced by THD throughtout the manuscript.

9 Comment 9: [The Study area and setting section is too lengthy. Be brief by removing all irrelevant and unrelated information.]

Response: Thank you for pointing this out, we have, revised this point.

10 Comment 10: [Under inclusion/exclusion, write the information in continuous sentences.]

Response: Thank you for pointing this out, we have, revised this point.

11 Comment 11: [11. Sample size determination. Your method looks quite confusing. Why don’t you use the cochrane formula once you know the prevalence of malaria from a previous study? If you are too strong about this method, show previous studies that used that method.]

Response: Thank you for pointing this out, the sample size calculation proportions of sensitivity and specificity method of Burderer and collaborators 1996, cited in Educational Notes in Research Methodology and Medical Statistics by Fahim et al., 2019: DOI: 10.22114/ajem.v0i0.158. Has been added in the manuscript

12 Comment 12: [Change Sampling technics to Sampling techniques.]

Response: Thank you for pointing this out, we have, revised this point.

13 Comment 13: [A drop of blood was placed in the centre the slide. State exact volume.]

Response: Thank you for pointing this out. A drop of 6 µL of blood was placed at the centre the slide. Has been stated in the manuscript

14 Comment 14: [Under thick film, please revise the information. It is not the applicator stick that hemolyses the samples and this is not why the sample is spread on the slide.]

Response: Thank you for pointing this out, we have, revised this point.

15 Comment 15: [Under thin film, state the exact volume as well.]

Response: Thank you for pointing this out. Approximately 3 µL of blood was placed on a clean labelled slide near its frosted end. we have, revised this point in the manuscript.

16 Comment 16: [Under microscopy, check the spelling of specie.]

Response: Thank you for pointing this out, “The thin films were used to identify the Plasmodium specie P. falciparum present”. Has been corrected.

17 Comment 17: [Procedure for RDT Test: Approximately 5μL of blood were used to diagnose malaria using the malaria antigen Plasmodium falciparum RDT kit, rephrase this statement. The volume of the blood is used to perform the test not used to diagnose malaria. Describe how the test was done and not the princples behind the test.]

Response: Thank you for pointing this out. “Approximately 5μL of blood were used to diagnose malaria perform the test using the malaria Ag P.f RDT kit, following the manufacturer’s instructions. The mRDT cassette was label with participant identification and 5ul of blood was applied to the sample pad on the test strip. Immediately after the blood clotted, the applicator was firmly applied in the circle and two drops of buffer were applied vertically above the circle and left for 10 minutes to migrate after which the results were read. The results were read following the manufacturer’s instructions as: if a single pink color appears on the control line “C”, the result was considered as negative. If a pink colored band appears on the “C” control line and a distinct pink colored band also appears on Pf line, the result was considered as positive.

The Ag P.f RDT kit is a diagnostic tool used to detect the presence of P. falciparum antigens in a patient's blood. This test is based on the detection of specific proteins produced by the malaria parasite, which are released into the bloodstream during an infection. When a small amount of blood was applied to the sample pad on the test strip, it flowed by capillary action through the strip and reached the area where the antigens are bound to the antibodies. If the P. falciparum antigens were present in the blood sample, they bound to the antibodies on the test strip, forming a visible coloured line. The RDT results were obtained in the field by the principal investigator, a certified and experienced laboratory scientist”. Has been revised in the manuscript

RESULTS

1 Comment 1: [Table 3: change NO to No. Same to YES.]

Response: Thank you for pointing this out, we have, revised this point.

2 Comment 2: [Define all the abbreviations under Validity of SD BIOLINE to diagnose P. falciparum and concurrence with Microscopy.]

Response: Thank you for pointing this out, FN = False Negative; TP= True Positive; FP= False Positive we have revised in the manuscript.

3 Comment 3: [Table of mRDT in diagnosis of P. falciparum compared to reference standard and studies in Cameroon. I did not see any previous study comparison.]

Response: Thank you for pointing this out, Table 5: Performance of mRDT in diagnosis of P. falciparum compared to reference standard and studies in Cameroon. Was a mistake, has been corrected

4 Comment 4: [Table 7, how do you classify parasite count as low, high, moderate etc. Justify these classifications with reference to a published paper.]

Response: Thank you for pointing this out, Parasite densities were graded using the ‘plus’ system scale: + (1 to 9 trophozoites in 100 fields); ++ (1 to 10 trophozoites in 10 fields); +++ (1 to 10 trophozoites per field); ++++ (>10 trophozoites per field) recommended by WHO. These scores were used to estimate parasite densities reported by Kosack et al [20] as low density = 10 to 90 parasites/μl; moderate density = 100 to 1,000 parasites/μl, and high density = ˃ 1,000 parasites/μl.

Has been revised under the methods section.

DISCUSSION

1 Comment 1: [The discussion section had your results repeated. Please discussion your findings in reference to malaria control in your country. Do not repeat the results.]

Response: Thank you for pointing this out, we have, revised this point.

2 Comment 2: [Be consistent with the use of HRP2, the discussion section has HRPII.]

Response: Thank you for pointing this out, HRPII has been replaced by HRP2 throughout the manuscript.

---

## [Editor Report · Decision Letter 2]

17 Sep 2024

PONE-D-23-43663R2Assessment of the diagnostic performance of the SD Bioline malaria antigen test for the diagnosis of malaria in the Tombel health district, Southwest region of CameroonPLOS ONE

Dear Dr. Watching Djakissam,

Thank you for submitting your manuscript to PLOS ONE. After careful consideration, we feel that it has merit but does not fully meet PLOS ONE’s publication criteria as it currently stands. Therefore, we invite you to submit a revised version of the manuscript that addresses the points raised during the review process.

Please note that the plus system scale is no longer recommended by WHO. Please revise this in the methodology and show the actual parasite counts in the results section.

We look forward to receiving your revised manuscript.

Kind regards,

Enoch Aninagyei, PhD

Academic Editor

PLOS ONE
---

## [Author Response · Author response to Decision Letter 3]

27 Oct 2024

Comments from Reviewer 1

1 Comment 1: [Please note that the plus system scale is no longer recommended by WHO. Please revise this in the methodology and show the actual parasite counts in the results section.]

Response: Thank you for pointing this out. We agree with this comment. Current WHO recommendation considered the use of density scale as low density = 10 to 90 parasites/μl; moderate density = 100 to 1,000 parasites/μl, and high density = ˃ 1,000 parasites/μl which is considered as the equivalent of plus system scale: + (1 to 9 trophozoites in 100 fields); ++ (1 to 10 trophozoites in 10 fields); +++ (1 to 10 trophozoites per field); ++++ (>10 trophozoites per field) recommended by WHO. These scores were used to estimate parasite densities reported by Kosack et al [20]. However, we removed this part from the manuscript

to have

Parasite densities were estimated according to parasite density levels reported by Kosack et al [20] as low density = 10 to 90 parasites/μl; moderate density = 100 to 1,000 parasites/μl, and high density = ˃ 1,000 parasites/μl.

comment from academic reviewer

Correct data availability statement

response: Data Availability The data generated or analyzed during the current study are available within the manuscript and its supportive documents

---

## [Editor Report · Decision Letter 3]

31 Oct 2024

Assessment of the diagnostic performance of the SD Bioline malaria antigen test for the diagnosis of malaria in the Tombel health district, Southwest region of Cameroon

PONE-D-23-43663R3

Dear Dr. Djakissam,

We’re pleased to inform you that your manuscript has been judged scientifically suitable for publication and will be formally accepted for publication once it meets all outstanding technical requirements.

Kind regards,

Enoch Aninagyei, PhD

Academic Editor

PLOS ONE
---

## [Editor Report · Acceptance letter]

PONE-D-23-43663R3

PLOS ONE

Dear Dr. Watching,

I'm pleased to inform you that your manuscript has been deemed suitable for publication in PLOS ONE. Congratulations! Your manuscript is now being handed over to our production team.

Kind regards,

on behalf of

Dr Enoch Aninagyei

Academic Editor

PLOS ONE